# High Levels of Policosanols and Phytosterols from Sugar Mill Waste by Subcritical Liquefied Dimethyl Ether

**DOI:** 10.3390/foods11192937

**Published:** 2022-09-20

**Authors:** Sudthida Kamchonemenukool, Chi-Tang Ho, Panatpong Boonnoun, Shiming Li, Min-Hsiung Pan, Wannaporn Klangpetch, Monthana Weerawatanakorn

**Affiliations:** 1Department of Agro-Industry, Faculty of Agriculture, Natural Resources and Environment, Naresuan University, Phitsanulok 65000, Thailand; 2Department of Food Science, Rutgers University, 65 Dudley Road, New Brunswick, NJ 08901, USA; 3Chemical Engineering Program, Department of Industrial Engineering, Faculty of Engineering, Naresuan University, Phitsanulok 65000, Thailand; 4College of Life Sciences, Huanggang Normal University, Huanggang 438000, China; 5Institute of Food Science and Technology, National Taiwan University, Taipei 10617, Taiwan; 6Faculty of Agro-Industry, Chiang Mai University, Chiang Mai 50100, Thailand; 7Cluster of High Value Products from Thai Rice and Plants for Health, Chiang Mai University, Chiang Mai 50100, Thailand

**Keywords:** subcritical extraction, filter mud, sugarcane leaves, bagasse, pretreatment

## Abstract

Extracting nutraceuticals with high value from bagasse, filter mud, and sugarcane leaves discarded as sugar mill by-products, is crucial for the development of a sustainable bio-economy. These by-products are important sources of policosanols and phytosterols, which have a cholesterol-lowering effect. This research focused on using a promising green technology, subcritical liquefied dimethyl ether extraction, with a low pressure of 0.8 MPa, to extract policosanols and phytosterols and on application of pretreatments to increase their contents. For direct extraction by subcritical liquefied dimethyl ether without sample pretreatment, the highest extraction yield (7.4%) and policosanol content were found in sugarcane leaves at 2888 mg/100 g, while the highest and lowest phytosterol contents were found in filter mud at 20,878.75 mg/100 g and sugarcane leaves at 10,147.75 mg/100 g, respectively. Pretreatment of filter mud by ultrasonication in hexane solution together with transesterification before the second subcritical liquefied dimethyl ether extraction successfully increased the policosanol content, with an extract purity of 60%, but failed to increase the phytosterol content.

## 1. Introduction

The FAO listed the area of sugarcane (*Saccarum officinarum* L.) at about 13 million ha, with a total commercial world cane production of about 1254.8 million ton/year [1]. Harvesting of sugarcane crops and processing at sugar mills generate significant amounts of by-products, such as sugarcane leaves, bagasse, and filter mud (filter cake) or pressed mud. Sugarcane leaves are agricultural waste from the harvesting step, whereas bagasse and filter mud are discarded during the milling process [2]. Harvesting methods by burning the leaves to facilitate sugarcane cutting and transportation to sugar factories [2] often cause environmental pollution that impacts human health, such as air pollution, with the PM_2.5_ concentration exceeding the guideline levels (less than 10 µg/m^3^ annual average). Adding value to sugarcane leaves as a source of high-priced nutraceuticals is a strategy that can sustainably solve the pollution problem caused by sugarcane burning.

Singh et al. [3] reported the occurrence of various nutraceuticals in sugarcane leaves and stalks, including policosanols and phytosterols. Both lipophilic molecules are stable compounds, possessing many bioactivities, including lowering blood cholesterol levels and reducing the risk of cardiovascular disease [4,5,6]. Policosanols comprise a mixture of long-chain aliphatic alcohols with chain lengths ranging from 22 to 34 carbons [7]. They are not predominantly available in regularly consumed plant foods. Major sources of policosanols are wax from sugarcane, rice bran, and honey from bees. Phytosterols, known as plant sterols, are found in plants, cereals, and nuts, especially seed oils. The most common phytosterols found in human diets are β-sitosterol, campesterol, and stigmasterol [8,9], with more than 200 types of phytosterols reported in plant species [10]. Feng et al. [11,12] reported that stigmasterol and β-sitosterol were the major phytosterols in sugarcane tips. 

Several techniques are used to extract nutraceuticals and the most widely used method is solvent extraction. The advantages of solvent extraction include convenience, ease of use, and hassle-free, while the disadvantages are the remaining chemical residues in the final product that are toxic to the environment [13]. Green technology, such as supercritical fluid extraction, has now been introduced as an environmentally friendly method, with no toxic residue in the final products. However, the use of high-pressure supercritical fluid technology is limited due to the cost [14]. Attard et al. [15] found policosanols and phytosterols in wax extracted by supercritical fluid carbon dioxide extraction at 35 MPa and 50 °C from sugarcane leaves and bagasse at 613.7, 913.6, 1087.8, and 275.1 mg/100 g, respectively [15], while Fernandes and Cabral [16] reported various extraction techniques for extraction and purification of phytosterols, including saponification, Soxhlet extraction with a solvent, esterification, transesterification, distillation, and supercritical fluid extraction (SFE) [11,16].

Subcritical liquefied extraction (SUBLE) is a new separation technology developed after supercritical fluid extraction (SFE). Both are a pressurized low-temperature technology, with similar advantages. Compared to SFE, the SUBLE technique requires lower pressure and costs less energy and investment, which makes it less expensive and more practical for large-scale applications [17,18]. Several solvents can be used as subcritical liquids, such as water, propane, butane, dimethyl ether (DME), and 1,1,1,2-tetrafluoroethane [17,18,19]. DME has been authorized by the European Food Safety Authority as a safe extraction solvent to produce foodstuffs and food ingredients [20]. The European Food Safety Authority (EFSA) authorized DME for food preparation of defatted animal protein products (EU Directive 2009/32/EC), and the Food Standards Australia New Zealand (FSANZ) [20] submitted DME as part of the applications to processing aid in the production of dairy and non-dairy food ingredients and products (FSANZ, 2011). DME is not a greenhouse gas and it does not deteriorate ozone layer [21,22,23]. Subcritical liquefied DME extraction (SUBLDME) has been used to extract various carotenoids and oils from various plants [24,25,26,27], while many studies reported higher yields of bioactive compounds extracted by SUBLE compared with SFE [28,29]. Our previous study demonstrated that SUBLDME extraction at a pressure of 0.8 MPa combined with transesterification offered the potential to recover policosanols from rice bran wax, a by-product of the rice bran oil process [30]. Our preliminary study also indicated that there were high contents of phytosterols extracted from sugarcane leaves by SUBLDME extraction. Consequently, this research aimed to apply the SUBLDME technique to extract policosanols and phytosterols from sugarcane leaves, bagasse, and filter mud, as sugarcane by-products. Soxhlet extraction with solvents was employed to compare the results. Furthermore, the influence of sample pretreatment before applying SUBLDME on the policosanol and phytosterol contents was investigated. The data suggest that SUBLDME is a promising green technique to extract policosanols and phytosterols, especially from the waste of sugar mills. This study adds significant value to processing sugar mill waste as sources of policosanols and phytosterols, using low-pressure subcritical dimethyl ether extraction technology.

## 2. Materials and Methods

### 2.1. Materials and Standard Reagents

The bagasse and filter mud were supplied by Phitsanulok Sugar Co., Ltd. (Pailom, Bangkratum Phitsanulok, Thailand), while sugarcane leaves at the harvesting stage were obtained from Rai-Hong-Kit-Jarean Organic Farm (Bantan, Chonnabot, Khon Kaen, Thailand). The samples were oven-dried at 60 °C for 24 h, then ground and sieved at the tested suitable size to yield the highest yield and bioactive contents. To determine the effect of sieving mesh sizes on yield in percentage and bioactive contents, defatted rice bran was used and sieved using the 40, 60, and 100 mesh sizes. All solvents and chemicals used were of analytical or gas chromatography (GC) grade. Ethanol, methanol, and 1,3,5-triphenylbenzene (TPB) were purchased from Sigma Chemical Co. (Beijing, China). DME was purchased from Siam Tamiya Co., Ltd. (Bangkok, Thailand).

All other solvents and chemicals at analytical or GC grade were purchased from RCI Labscan (Bangkok, Thailand). Policosanol standards, including docosanol (C-22), tetracosanol (C-24), hexacosanol (C-26), octacosanol (C-28), triacontanol (C-30), dotriacontanol (C-32), tetratiacontanol (C-34), and phytosterol standards, including campesterol, stigmasterol, β-sitosterol, sitostanol, and 5α-cholestane, were purchased from Sigma-Aldrich (St. Louis, MO, USA). *N*,*O*-Bis(trimethylsilyl)-trifluoroacetamide (BSTFA) with 1% trimethylchlorosilane (TMCS) and pyrogallol were sourced from Sigma-Aldrich.

### 2.2. Solvent Extraction of Crude Extract from Samples 

Approximately 10 g of samples were extracted using the Soxhlet method in cellulose thimbles (Whatman 33 mm × 100 mm) with a 280 mL mixture of hexane and methanol (20:1 *v*/*v*) for 18 h. A rotary evaporator under vacuum condition at 40 °C was applied to remove the solvents from the extract. The obtained samples were kept at −20 °C until required for further analysis. For policosanol analysis, the dry extract was dissolved with 4 mL toluene, filtered using a syringe filter with 0.45 µm pores, and then transferred into a vial for later analysis [31]. 

### 2.3. Extraction from By-Product Samples by SUBLE Using DME 

A lab-scale apparatus of SUBLDME with a capacity of 10 g was used to extract the crude extract containing policosanols and phytosterols from the samples, including sugarcane leaves, bagasse, and filter mud. A schematic illustration is shown in Figure 1. The sample was placed in a cellulose thimble (Whatman 30 mm × 100 mm) and a known amount of DME was transferred into a stainless-steel batch extractor that was heated by a hot plate stirrer. To determine the effect of temperature on bioactive contents, the reactor temperature was set to low (35 °C) and high (60 °C) levels by thermocouple. The pressure was set at 0.8 MPa to liquefy the DME, and the extraction time (30 min) and stirring speed by a magnetic stirrer bar were also set at one condition. The extracted sample was passed through a metal filter (7 µM) and stored at −20 °C until further application. The obtained extract was processed to analyze yield percentage, policosanol and phytosterol compositions.

### 2.4. Policosanol Analysis

#### 2.4.1. Policosanol Derivatization

The extracted sample (1 mL) was spiked with 25 µL of cholestane as the internal standard (2 mg/mL) and kept in a glass vial for further analysis. Trimethylsilyl (TMS) derivatives of the policosanols were prepared using a modified method from Asikin, Chinen, Takara, and Wada [31]. *N*-Methyl-*N*-(trimethylsilyl) trifluoroacetamide reagent (200 µL) was added to the sample extracts (100 µL) and an internal standard mixture solution. After mixing, the solutions were heated at 50 °C for 30 min. 

#### 2.4.2. Gas Chromatography–Mass Spectrometry (GC-MS) Conditions for Policosanol Analysis

Policosanol analyses were carried out after derivatization using a GC-MS (Shimadzu GC-2010 plus QP 2010, Kyoto, Japan) equipped with an Agilent J&W Scientific DB-5ms fused silica capillary column (0.25 mm × 30 m × film thickness 0.25 µm). The samples (1 µL) were directly injected onto a GC-MS column with a split ratio of 1:10. The GC oven temperature was programmed from 150 (holding time 2 min) to 320 °C at a rate of 4 °C/min and then maintained at 320 °C for 15 min. Injector and transfer line temperatures were set at 280 °C. The electron impact (EI) ion source temperature was set to 200 °C and the ionization energy was set to 70 eV; a single ion monitoring mode (SIM) was set to identify and quantify trimethylsilylated (TMS) policosanols. Docosanol (C22), tetracosanol (C24), hexacosanol (C26), octacosanol (C28), triacontanol (C30), dotriacontanol (C-32), and tetratriacontanol (C-34) identified and quantified according to their retention times and molecular target ion, *m*/*z* at 374, 383, 411, 439, 467, 495, and 524, respectively [32]. 

### 2.5. Analysis of Phytosterols

#### 2.5.1. Extraction of Phytosterols

The extraction was conducted based on the methods of Beveridge et al. [33] and Thanh et al. [34] with some modifications. A sample (1 g) was accurately weighed into a screw tube and spiked with 1 mL of internal standard into the matrix, followed by 2 mL of aqueous KOH (60%), 2 mL of ethanol (95%), 2 mL of aqueous NaCl (10%), and 0.3 g of pyrogallol (antioxidant). The mixture was saponified under nitrogen (N_2_) and then incubated at 70 °C for 45 min with vortexing every 15 min to ensure complete saponification. After the solution was cooled in an ice bath, the saponified portion was extracted with 15 mL of hexane and ethyl acetate (9:1, *v*/*v*) twice. The upper layer (unsaponifiables) was collected into a glass tube and the solvent evaporated to dryness under a stream of N_2_ at 45 °C.

#### 2.5.2. Phytosterol Derivatizations

Phytosterol derivatizations were prepared following a method modified from Beveridge et al. [33]. The residue was mixed with 200 µL of *N*,*O*-bis(trimethylsilyl) trifluoroacetamide (BSTFA) with 1% trimethylchlorosilane (TMCS) and 100 µL of pyridine, then heated at 60 °C for 30 min and evaporated to dryness under a stream of N_2_ at 45 °C. The residue was dissolved in 1 mL hexane and the solution was filtered using a syringe filter with 0.45 μm pores then transferred into a vial for GC-MS analysis.

#### 2.5.3. Quantification and Identification of Phytosterols by GC-MS 

Quantification and identification of sterols were performed by GC-MS following the modified method of Thanh et al. [34]. The fused silica capillary GC column was a HP-5MS (30 m × 0.25 mm i.d. × film thickness 0.25 µm). The samples (1 µL) were injected using an Agilent Technologies 7683 Autosampler and a split injector with a split ratio of 1:50. The sterols were separated using programmed oven temperatures originally set at 100 °C (1 min), then raised to 300 °C (14 min) at a rate of 10 °C/min. Helium was used as a carrier gas at a flow rate of 1.5 mL/min. The ionization energy was set to 70 eV. The injector, MS quad temperatures, MS source, and transfer lines were 270, 150, 230, and 280 °C, respectively. TMS-phytosterols were identified and quantified by a SIM (single ion monitoring) mode according to their retention times and MS spectra. The *m*/*z* ratios of the ions used for quantitative analysis were TMS-campesterol (472), TMS-stigmasterol (484), TMS-β-sitosterol (486), and TMS-sitostanol (488) [34]. 

### 2.6. Sample Pretreatment before SUBLDME Extraction on Policosanol and Phytosterol Contents

The filter mud was used to increase the policosanol and phytosterol contents. The filter mud was divided into four parts to replicate four conditions, including the control group. Sample pretreatment conditions included sonication and transesterification, as shown in Table 1. The control was filter mud extracted by SUBLDME without pretreatment. The remaining three conditions were applied as ultrasonic treatment in hexane before extraction using SUBLDME, transesterification, and the second extraction with a subcritical technique using DME. Ultrasonic pretreatment in hexane was performed by mixing the sample in hexane (1:4 *w*/*v*) and subjection to ultrasonic treatment at a 35 kHz frequency (GT-1730 QTS, Guangdong, China) at 100 W for 30 min. The mixture was then allowed to stand at room temperature for 24 h until a dried sample was obtained. Pretreatment by transesterification followed the method of Wongwaiwech et al. [30] was applied. Briefly, the sample was dissolved in a solution of NaOH in EtOH (2%) with stirring at 80 °C. The solution was allowed to react and then mixed with a warm solution of isooctane and EtOH (70%:30%). The isooctane layer was separated and kept at 4 °C overnight. The crystallized wax formed was filtered using a Buchner funnel and washed twice with EtOH. The precipitate was kept and dried in a hot air oven at 60 °C. The dried extract was then ground and kept at −20 °C until required for other applications. 

### 2.7. Statistical Analysis

The results are expressed as the means ± standard deviation (SD) based on dry weight for the triplicate analyses on the same sample. Analysis of variance (ANOVA) was performed to analyze the data with Duncan’s tests using SPSS 19 software (SPSS Inc., Chicago, IL, USA). The significant difference level was set at *p* < 0.05. 

## 3. Results and Discussion

### 3.1. Effect of Particle Size and Extraction Temperature on Yield and Phytosterol Content by SUBLDME Extraction

Sieving mesh sizes of defatted rice bran at 40, 60, and 100 mesh and extraction temperatures of 35 and 60 °C were investigated to determine whether these factors impacted extraction yields and bioactive contents by SUBLDME technique. The yields and phytosterol contents are shown in Table 2. The 100 mesh sample size and extraction temperature of 60 °C resulted in the highest extraction yield at 2.68%. The results indicated that a higher extraction temperature and smaller particle size of the samples (100 mesh) produced a higher yield of crude extract. The highest phytosterol content was also found at 5476.7 mg/100 g with a sieving mesh size of 100 at 60 °C, while the phytosterol content (5404.25 mg/100 g) using a sieving mesh size of 60 and extraction temperature of 60 °C was not significantly different (*p* < 0.05). Therefore, the sieving mesh size of 60 and extraction temperature of 60 °C were applied in all further experiments. Many factors affected the yields and extracted bioactive content, with the most significant parameters impacting extraction efficiency, in terms of yield and quantity, being temperature, time, and particle size, where too fine or too coarse sample particles decreased the extraction efficiency [35,36,37,38]. Optimum particle size should be determined by preliminary studies to obtain the maximum extraction yield. Previous studies reported similar trends regarding the effect of particle size on the extraction yields and antioxidant activities of peanut skin. A particle size of 425 μm yielded the highest oil yield compared to a higher particle size of 500 μm and smaller particle size of 355 μm [39]. A decrease in particle size resulted in a greater surface area of particles in contact with the solvent, enhancing the leaching of active compounds due to increased mass transfer. Moreover, a decrease in particle size and gradual increase in temperature also positively affected the extraction rate [40,41]. 

### 3.2. SUBLDME Extraction of Policosanols and Phytosterols Compared with Solvent on Soxhlet Extraction

Soxhlet extraction with a solvent mixture of hexane and methanol was successfully used to extract wax from sugarcane for policosanol analysis [13,31]. Consequently, SUBLDME extraction of sugarcane by-products, including bagasse, filter mud, and sugarcane leaves, was performed, and the results were compared with a solvent mixture of hexane and methanol using Soxhlet extraction. The obtained crude extract was processed to analyze the policosanol and phytosterol contents by GC-MS. The yield percentages of the crude extract and policosanol and phytosterol contents are shown in Table 3. The GC-MS chromatograms of the sugarcane leaves and standard policosanols are shown in Figure 2 and Figure 3. Since each sample had a different food matrix and components, the same method of extraction gave diverse results. Compared with the solvent after Soxhlet extraction, SUBLDME gave high yield percentages for all samples except bagasse. The highest yield was found at 7.04% (cane leaves) by SUBLDME extraction and the lowest yield at 1.22% (filter mud) by the solvent after Soxhlet extraction. Compared with bagasse and filter mud, sugarcane leaves provided the highest yield percentage from both SUBLDME (7.04%) and solvent extraction (3.59%). Attard et al. [15] also found that supercritical fluid CO_2_ extraction of sugarcane leaves gave the highest yield (1.6%) from the crude extract compared with bagasse and rind. Bagasse, similar in components to wood, is a fibrous material consisting of 45% cellulose, 28% hemicellulose, 20% lignin, 5% sugar, 1% minerals, and 2% ash [42], which is linked to being an obstacle for liquefied DME to diffuse to release the bioactive compounds.

The results also showed that both the policosanol and phytosterol contents of crude extracts from all the by-products by SUBLDME extraction were significantly higher than the solvent after Soxhlet extraction (Table 3). With the highest extraction yield, sugarcane leaves extracted by SUBLDME provided the highest policosanol content (2845.71 mg/100 g) followed by bagasse (2446.82 mg/100 g). Although the bagasse gave the lowest yield percentage (2.95%) compared with filter mud (4.26%), the policosanols contents of bagasse (2444.29 mg/100 g) were remarkably higher than that of filter mud (668.90 mg/100 g) by SUBLDME. For an efficient extraction process, it is important that the cell wall is permeable and can be extensively disrupted. Plant tissue is generally compartmentalized. Different bioactive components are located in different parts of the cellular tissue and the tissue structure plays an important role in the release of chemical components [43]. For the same extraction method using different samples, the extraction yield and the released bioactive contents from solid–liquid extraction depend on many parameters. The crucial factor is the solid matrix of samples, especially the microstructural effect. The different solid matrices of sugarcane leaves, bagasse, and filter mud caused diverse diffusivity as a result of structural complexity impacts such as tortuosity, porosity, and volume fraction [43,44]. By fluorescence microscopy (OLYMPUS, Model BX53F2, Tokyo, Japan), the outermost layers of the sugarcane leaves after SUBLDME were seen to be slightly damaged and displayed wrinkles intermittently (Appendix A). There was a rougher surface and some round particles that were exposed on the surface of it, indicating that the surface was significantly damaged, and the round particles almost disappeared. Bao et al. [45] found a similar result after extraction of lotus seedpods by gas-assisted, glycerol extraction [45].

For supercritical fluid CO_2_ extraction (SF-CO_2_) at 35 MPa and 50 °C, Attard et al. [15] found the total policosanol content of sugarcane leaves and bagasse at 613.7 and 913.6 mg/100 g, respectively. The policosanol contents found in this study were much higher, at 4.5 and 2.6 times, for sugarcane leaves and bagasse, respectively. Therefore, SUBLDME extraction with a low pressure (0.8 MPa at 60 °C for 30 min) showed the potential for extraction of policosanols. The main policosanols found were octacosanol (C-28) and triacontanol (C-30), consistent with other studies where octacosanol (C-28) and triacontanol (C-30) were the predominant policosanols found in sugarcane [15,46,47]. The filter mud by SUBLDME technique gave the lowest total policosanol content (668.90 mg/100 g), with an octacosanol (C-28) level of 249.21 mg/100 g. Ou et al. [48] used SF-CO_2_ (30 MPa at 50 °C for 2 h) with ethanol (500 mL) as a co-solvent to extract octacosanol (C-28) from filter mud. They found the highest yield of crude wax at 5.51%, with a high content of octacosanol (C-28), at 29,650 mg/100 g. Compared with SF-CO_2_, Ou et al. [48] found a lower extraction yield and policosanol content in filter mud, suggesting that SF-CO_2_ with ethanol as co-solvent was more suitable for extraction of policosanols from filter mud than SUBLDME extraction. The ethanol used in Ou et al.’s study increased the polarity of the carbon dioxide fluid and the solvating power towards obtaining the bioactive compounds. 

Interestingly, the phytosterol contents by SUBLDME method were dramatically high for all samples. The highest phytosterol contents were found in the filter mud (20,878.75 mg/100 g) followed by the bagasse (19,556.33 mg/100 g). The GC-MS chromatograms of the filter mud and standard phytosterols are shown in Figure 4 and Figure 5. These results strongly contrast with the data of the policosanol contents, for which the lowest and highest were found in filter mud and sugarcane leaves, respectively. Attard et al. [15] found the phytosterol contents in crude extract of sugarcane leaves and bagasse at 1087.8 and 275.1 mg/100 g, respectively, using SF-CO_2_ (35 MPa at 50 °C). This study found the lowest amount (10,147.55 mg/100 g) of phytosterols in the sugarcane leaves by SUBLDME extraction (0.8 MPa at 60 °C). However, the lowest amount of phytosterols (10,147.55 mg/100 g) in the sugarcane leaves was still 9 times higher than the one found by Attard et al. [15], which was at 1087.8 mg/100 g. 

SUBLDME extraction is used to explain DME at a temperature and pressure below the critical temperature (Tc) (126.9 °C) and pressure (Pc) (5.3 MP), as well as adequately over ambient conditions [49]. DME exists in a gaseous state at atmospheric conditions (0.1 MPa at 25 °C) and is liquefied at a pressure of more than 0.51 MPa (vapor pressure) at room temperature. The liquefied state has a low viscosity, low density, and high solubility in water, or a high degree of miscibility with water, leading to high diffusivity compared with CO_2_ [21,50,51]. The density of liquefied DME (1.0 MPa at 60 °C) was about 600 kg/m^3^ whereas fluid CO_2_ was estimated at around 200–900 kg/m^3^ based on temperature and pressure [48]. The viscosity of supercritical CO_2_ fluid at 0.1–0.3 MPa [52] was also high compared with liquefied DME (less than 0.15 MPa). These physical properties are similar; however, the relative permittivity (ε_r_), one of the indicators related to the degree of substance polarity of DME, was higher than CO_2_ [53], suggesting the polarity of phytosterol being closer to the extraction conditions of subcritical liquefied DME compared with fluid CO_2_. These data imply that organic co-solvents, such as ethanol, could be applied to increase the phytosterol content by SF-CO_2_, since the solvating power of fluid CO_2_ increased with addition of organic co-solvents [54]. Compared with SUBLDME extraction, the very low phytosterol contents by SF-CO_2_ shown by Attard et al. [15] might be due to no co-solvent applied in the extraction system. The supercritical condition might lead to a lower solvating power compared with the SUBLDME technique used in this study. The sugarcane leaves were determined as an excellent source of phytosterols for SFE-CO_2_ extraction by Attard et al. [15]. Among the extraction parameters, the temperature and flow rate had strongly affected the phytosterol content and there was no effect of pressure on the phytosterol content by SF-CO_2_ without a co-solvent [55]. Here, we found that the filter mud and the bagasse were excellent sources of phytosterols (20,878.75 mg/100 g and 19,556.33 mg/100 g, respectively), while the leaves were a good source of policosanols by SUBLDME extraction. 

Among the phytosterols of the crude extracts, β-sitosterol was the most prominent, both from solvent extraction and SUBLDME, accounting for 41–60% of the total phytosterols, followed by stigmasterol (21–40%) and campesterol. β-Sitosterol was the major phytosterol found in sugarcane leaves and bagasse (57% and 42%, respectively), followed by stigmasterol [11,12,15,56]. From a large amount of evidence, both in vitro and in vivo, Babu and Jayaraman [8] reported that besides the cholesterol-lowering effect, β-sitosterol showed high potential as a herbal nutraceutical drug of the future for diabetic treatment [8].

The results indicate SUBLDME extraction to be a promising green technology, with low-pressure extraction of policosanols and phytosterols from filter mud, bagasse, and sugarcane leaves. José et al. [57] reported that the air pollution from sugarcane burning in 1997–1998 affected human respiratory systems, especially children and the elderly, with an increase in PM_2.5_ to 10.2 µg/m^3^ and PM_10_ to 42.9 µg/m^3^, leading to higher incidences of respiratory diseases. The results showed that sugarcane leaves and filter mud have the potential to create added value as sources of policosanols and phytosterols, respectively, and offer a sustainable solution to reduce pollution caused by sugarcane burning. However, further studies comparing these two green technologies (subcritical and supercritical extraction methods) for extraction and purifying without degradation or decomposition of bio-compounds are required.

### 3.3. Effect of Sample Pretreatment following SUBLDME Extraction on the Yield and Policosanol and Phytosterol Contents of Filter Mud 

Sample pretreatment was investigated to increase both nutraceuticals. The filter mud had the highest amount of phytosterols and the lowest amount of policosanols by SUBLDME extraction. Therefore, the filter mud was used as a sample to determine the effects of sample pretreatment on the amounts of these lipophilic nutraceuticals. Since this study focused on policosanols and phytosterols naturally found in plant wax, hexane was used in the ultrasonic pretreatment. The filter mud directly extracted by SUBLDME extraction without any pretreatment was the control, as condition number 1 in Table 1. Sample pretreatment by ultrasonication in hexane, followed by extraction by SUBLDME extraction, was condition number 2. The crude extract obtained from condition 2 and subjected to transesterification was placed in condition number 3, while the obtained extract from condition 3 subsequently extracted by SUBLDME was set as condition 4. Yield percentage and the policosanol and phytosterol contents are reported in Table 4.

The sample pretreatments, including ultrasonication and transesterification, following SUBLDME extraction reduced the yield of the extraction process from 4.19% to 1.55% based on the filter mud sample. The results showed that the extraction yield in condition 3 was 18.9% based on the obtained extract from condition 2, while condition 4 was 30.6% based on the obtained extract from condition 3 (data not shown). The filter mud contained sugarcane crude wax of around 55–62% by heptane extraction [58]. They also recorded the extraction yield from the filter mud at 3.5% from ultrasound-assisted extraction (1:7 *w/v* for 3 h) by hexane. This study showed that the same wax extraction process with a small amount of hexane and shorter extraction time (1:4 *w/v* for 30 min) together with SUBLDME extraction gave a slightly higher yield (4.75%). Variations in results occurred because of different sample origins (Thailand and Cuba), varieties of sugarcane, and agricultural processes.

The ultrasonic treatment in hexane followed by the second extraction with SUBLDME dramatically increased the content of policosanols from 668.90 to 5589.87 mg/100 g, and then the transesterification process slightly increased the policosanol content to 6317.00 mg/100 g. However, policosanol contents dramatically increased to 60,056.2 mg/100 g after applying the second SUBLDME extraction to the transesterified extract. The results indicated that the policosanol content of the extract increased from 668.90 (control condition) to 60,056.23 mg/100 g after all the pretreatment processes, which is a 90-fold increase from the control sample. Our previous report indicated that pretreatment of rice bran wax discarded from the rice bran oil industry by transesterification (without Soxhlet extraction) increased the policosanol content from 332.79 mg/100 g to 30,787.89 mg/100 g. The application of SUBLDME extraction then increased the policosanol content of the obtained transesterified wax from 30,787.89 to 84,913.14 mg/100 g [30]. Using the SF-CO_2_ method with ethanol as co-solvent, filter mud gave a policosanol content of 1297 mg/100 g [48], lower than our results. De Lucas et al. [47] used pretreatment of sugar wax (55.2% wax) by saponification before application of SF-CO_2_ and found that the highest yield obtained was 1.9%, with the highest total content of policosanols at 78 mg/100 g (C22-C34) at a pressure of 35 MPa and temperature of 100 °C. This result indicated that pretreatment of filter mud by ultrasonication in hexane solution together with transesterification before the second extraction with SUBLDME successfully increased the policosanol content, giving an extract purity of 60%. 

The effect of sample pretreatment for the phytosterol content in the crude extract differed from the policosanol content. The phytosterol content of the extract decreased from 20,878.75 mg/100 g (control condition) to 9998.18 mg/100 g after applying all the pretreatment processes. The ultrasonic treatment with hexane and transesterification did not increase the phytosterol content. Ultrasound treatment with hexane destroyed the biological cells in the food matrix and degraded the phytosterols without any harmful effects on the policosanol content. Consequently, high amounts of policosanols were released into the extract compared with phytosterols. The result suggests that the polarity of the policosanols was closer to hexane compared with phytosterol compounds. Dunford et al. [59] showed that the high temperatures (60–100 °C) required to extract the phytosterols from wheat straw and hexane extraction at 80 °C (10 MPa) gave the lowest amount of phytosterols compared with petroleum ether, chloroform, and ethanol. In this study, the phytosterols were hard to release, as no heating was applied during the ultrasonic treatment. After the ultrasonic treatment, following extraction by SUBLDME, transesterification dramatically reduced the phytosterol content. In plant matrices, sterols are naturally present in ester form. The process of hydrolysis converts phytosterols to a free form and saponification is required before extraction [60]. Esterification and transesterification result in the synthesis of plant sterol esters, especially in the presence of acid catalysts [61]. However, the second extraction by SUBLDME increased the phytosterols from 4655.90 to 9998.18 mg/100 g, indicating that extraction by SUBLDME showed promising potential to extract phytosterols, while ultrasonication with hexane and transesterification was not suitable for the pretreatment of filter mud before SUBLDME extraction. Saponification and microwave-assisted extraction were reported for successive extraction of phytosterols [60,62]. Other sample pretreatment techniques, such as saponification and microwave-assisted extraction, prior to phytosterol extraction with SUBLDME should be investigated in future studies.

## 4. Conclusions

Subcritical liquefied dimethyl ether extraction of policosanols and phytosterols from the by-products of sugar mills, including bagasse, filter mud, and sugarcane leaves, was successfully performed to obtain high-value nutraceutical compounds. Extraction of different by-products resulted in diverse yields and nutraceutical contents. Filter mud provided an excellent source of phytosterols (20,878 mg/100 g), while sugarcane leaves generated a substantially higher content of policosanols (2845 mg/100 g). Pretreatment of filter mud by ultrasonication in hexane solution, together with transesterification before the second extraction with SUBLDME, increased the policosanol but not phytosterol content. To increase the phytosterol content, other pretreatment techniques, including saponification, microwave-assisted extraction, and other extraction processes at high temperature before extraction of SUBLDME, are required for further investigation. 

## Figures and Tables

**Figure 1 foods-11-02937-f001:**
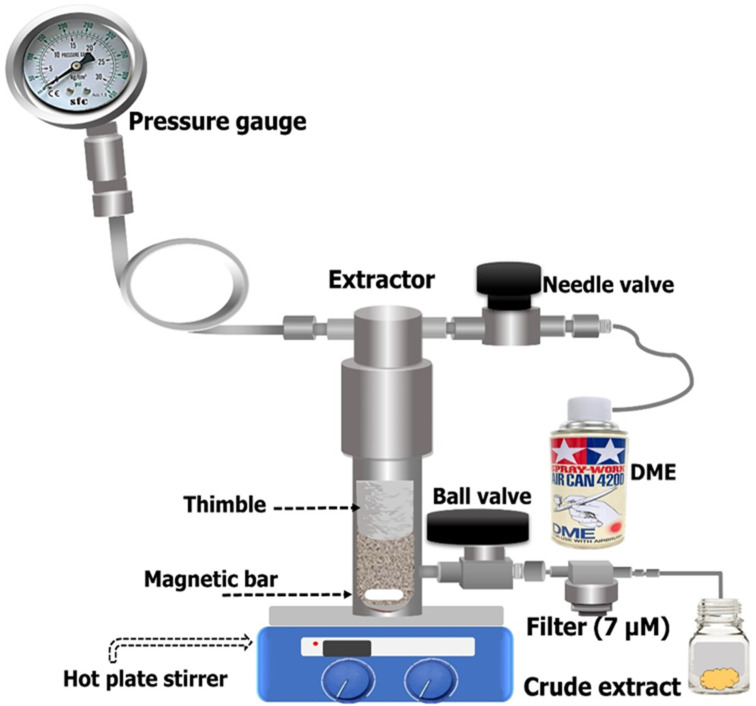
A schematic illustration of the subcritical liquefied DME extraction process.

**Figure 2 foods-11-02937-f002:**
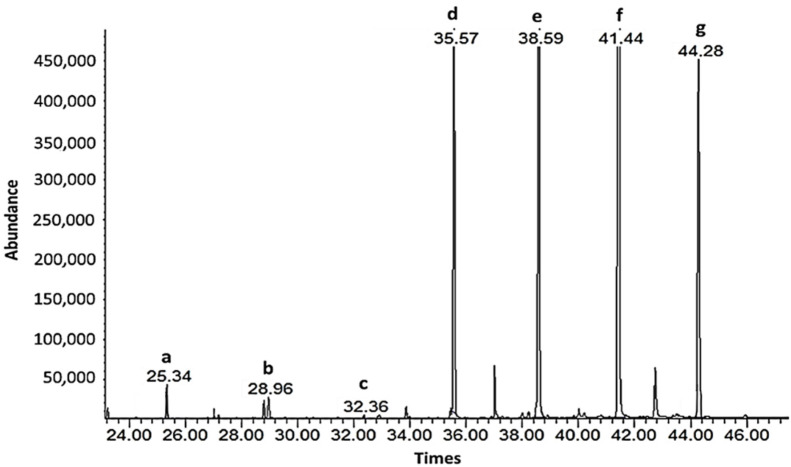
GC-MS chromatograms of the policosanol contents from sugar cane leaves: a. Docosanol (C-22); b. Tetracosanol (C-24); c. Hexacosanol (C-26); d. Octacosanol (C-28); e. Triacosanol (C-30); f. Dotriacontanol (C-32); g. Tetratriacontanol (C-34).

**Figure 3 foods-11-02937-f003:**
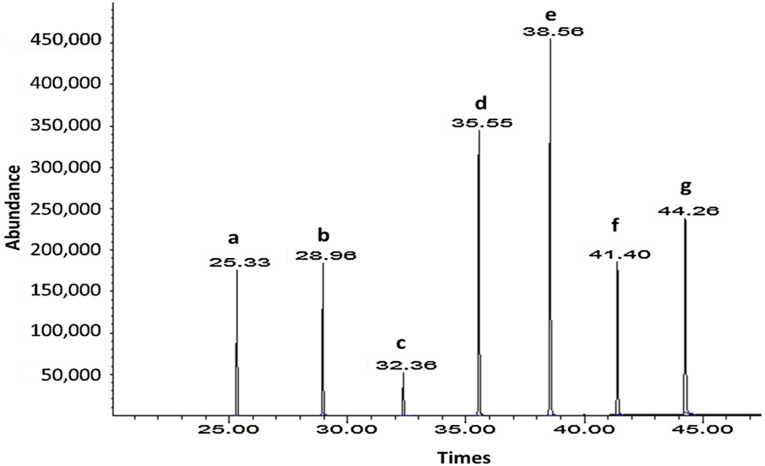
GC-MS chromatograms of standard policosanol at 100 ppm: a. Docosanol (C-22); b. Tetracosanol (C-24); c. Hexacosanol (C-26); d. Octacosanol (C-28); e. Triacosanol (C-30); f. Dotriacontanol (C-32); g. Tetratriacontanol (C-34).

**Figure 4 foods-11-02937-f004:**
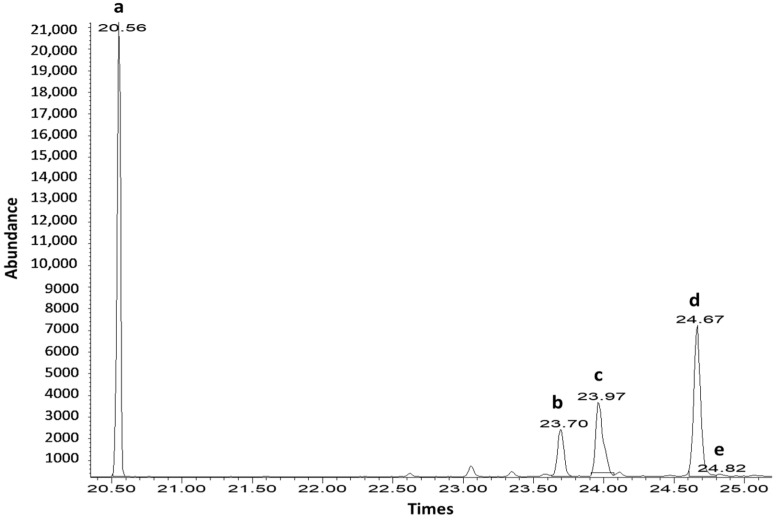
GC-MS chromatograms of the phytosterol contents from filter mud. The numbers on the peaks correspond to (a) cholestane as internal standard, (b) campesterol, (c) stigmasterol, (d) b-sitosterol, and (e) sitostanol.

**Figure 5 foods-11-02937-f005:**
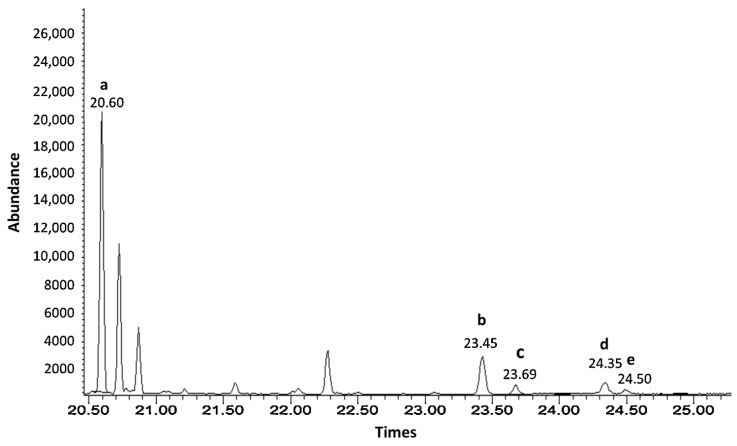
GC-MS chromatograms of the standard phytosterol at 50 ppm. The numbers on the peaks correspond to (a) cholestane as internal standard, (b) campesterol, (c) stigmasterol, (d) b-sitosterol, and (e) sitostanol.

**Table 1 foods-11-02937-t001:** Conditions for policosanol extraction of filter mud.

Conditions	Ultrasonic Treatment with Hexane	SubcriticalLiquefied DME	Transesterification	SubcriticalLiquefied DME
1	**-**	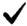	-	**-**
2	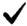	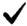	-	**-**
3	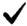	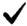	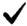	**-**
4	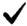	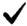	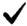	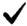

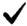
 = The used process in each condition.

**Table 2 foods-11-02937-t002:** Phytosterol contents of defatted rice bran extracted by SUBLDME with different sample sizes and extraction temperatures.

Size/Temp (Mesh/°C)	%Yield	Phytosterol Contents (mg/100 g)				Total
		Campesterol	Stigmasterol	Beta-Sitosterol	Sitostanol	
40/35	2.14 ± 0.17 ^e^	884.49 ± 30.60 ^b^	507.54 ± 2.11 ^b^	787.80 ± 27.78 ^e^	145.84 ± 23.70 ^c^	2325.69 ± 32.58 ^d^
40/60	2.32 ± 0.76 ^b^	1381.7 ± 22.34 ^a, b^	689.00 ± 35.83 ^b^	1252.54 ± 3.52 ^d^	286.49 ± 40.15 ^b, c^	3609.73 ± 21.54 ^c^
60/35	2.04 ± 0.37 ^f^	1576.28 ± 3.60 ^a, b^	883.82 ± 6.24 ^a, b^	1376.51 ± 22.33 ^c, d^	373.53 ± 28.03 ^a, b^	4210.14 ± 60.21 ^b, c^
60/60	2.18 ± 0.14 ^d^	1619.47 ± 773.40 ^a, b^	1513.24 ± 563.18 ^a^	1931.06 ± 264.90 ^b^	340.47 ± 121.22 ^a, b^	5404.25 ± 826.73 ^a^
100/35	2.28 ± 0.27 ^c^	659.17 ± 15.68 ^b^	1655.97 ± 28.01 ^a^	2492.13 ± 16.84 ^a^	299.68 ± 0.34 ^a, b, c^	5106.96 ± 4.16 ^a, b^
100/60	2.68 ± 0.38 ^a^	2113.67 ± 19.38 ^a^	1180.95 ± 8.38 ^a, b^	1707.73 ± 61.18 ^b, c^	474.25 ± 43.85 ^a^	5476.70 ± 132.78 ^a^

Each value represents the mean ± SD. Values with different superscript letters in the same column are significantly different (*p* < 0.05). Mesh = sieving size of defatted rice bran, from 40 to 100 mesh.

**Table 3 foods-11-02937-t003:** Policosanol and phytosterol contents of by-products using SUBFE with DME and Soxhlet extraction with a solvent.

	Soxhlet Extraction			SUBFE with DME Extraction		
	Bagasse	Leaves	Filter Mud	Bagasse	Leaves	Filter Mud
% Yield	3.49 ± 0.08 ^b^	3.59 ± 0.32 ^a^	1.22 ± 0.31 ^d^	2.95 ± 0.12 ^c^	7.04 ± 0.22 ^a^	4.26 ± 0.02 ^b^
Policosanol contents (mg/100 g)						
C22	3.70 ± 0.11 ^d, A^	1.97 ± 0.08 ^e, B^	0.90 ± 0.25 ^e, C^	7.82 ± 0.26 ^c, B^	9.45 ± 1.36 ^b, B^	16.87 ± 0.20 ^a, A^
C24	12.83 ± 0.21 ^c, A^	1.16 ± 0.20 ^d, B^	0.15 ± 0.04 ^d, C^	20.52 ± 2.40 ^b, B^	35.25 ± 0.16 ^a, A^	23.85 ± 5.24 ^b, B^
C26	14.05 ± 0.03 ^c, A^	0.63 ± 0.16 ^e, B^	1.40 ± 0.39 ^e, B^	129.41 ± 1.77 ^a, A^	113.43 ± 0.08 ^b, B^	8.47 ± 2.38 ^d, C^
C28	163.54 ± 16.21 ^d, A^	12.43 ± 1.26 ^e, B^	30.94 ± 8.60 ^e, B^	1737.10 ± 2.42 ^b, B^	2072.96 ± 1.08 ^a, A^	249.21 ± 1.39 ^c, C^
C30	18.36 ± 1.19 ^d, A^	13.26 ± 3.47 ^e, A^	5.28 ± 1.46 ^f, B^	328.79 ± 0.82 ^b, B^	411.89 ± 0.82 ^a, A^	102.72 ± 8.03 ^c, C^
C32	91.32 ± 0.54 ^d, A^	32.32 ± 0.12 ^e, B^	4.53 ± 1.25 ^f, C^	184.07 ± 3.71 ^b, B^	177.14 ± 0.28 ^c, B^	238.28 ± 4.85 ^a, A^
C34	50.32 ± 0.47 ^a, A^	10.77 ± 3.10 ^d, B^	0.60 ± 0.17 ^e, C^	39.11 ± 2.71 ^b, A^	25.61 ± 2.17 ^c, B^	29.50 ± 2.22 ^c, B^
Total	354.12 ± 16.33 ^d, A^	72.54 ± 8.14 ^e, B^	43.78 ± 12.16 ^e, B^	2446.82 ± 14.07 ^b, B^	2845.71 ± 1.45 ^a, A^	668.90 ± 19.57 ^c, C^
Phytosterol contents (mg/100 g)						
Campesterol	8.70 ± 1.25 ^d, B^	11.12 ± 3.01 ^d, B^	37.17 ± 1.92 ^d, A^	3788.98 ± 17.62 ^b, B^	966.83 ± 33.94 ^c, C^	4482.19 ± 1.84 ^a, A^
Stigmasterol	18.16 ± 0.02 ^d, C^	35.00 ± 3.83 ^d, B^	101.32 ± 0.06 ^d, A^	7234.98 ± 37.46 ^a, A^	4014.32 ± 112.47 ^c, C^	4423.49 ± 14.95 ^b, B^
Beta-sitosterol	47.68 ± 1.86 ^e, C^	65.00 ± 6.78 ^e, B^	137.18 ± 1.24 ^d, A^	8078.10 ± 46.41 ^b, B^	4848.16 ± 11.71 ^c, C^	11,804.11 ± 7.09 ^a, A^
Sitostanol	4.16 ± 0.31 ^d, B^	14.69 ± 3.56 ^d, A^	11.70 ± 0.75 ^d, A^	454.28 ± 2.89 ^a, A^	318.24 ± 21.30 ^b, B^	168.92 ± 6.48 ^c, C^
Total	78.67 ± 0.28 ^e, C^	125.81 ± 11.15 ^e, B^	287.36 ± 4.00 ^d, A^	19,556.33 ± 104.38 ^b, B^	10,147.55 ± 113.41 ^c, C^	20,878.75 ± 17.41 ^a, A^

Each value represents the mean ± SD. Values with different superscript lowercase letters in the same row are significantly different (*p* < 0.05). Values with different superscript uppercase letters in the same row and the same extraction method are significantly different (*p* < 0.05).

**Table 4 foods-11-02937-t004:** Policosanol and phytosterol contents of filter mud with different pretreatments before extraction by SUBFE with DME.

	Extraction Conditions			
	1	2	3	4
Percent yield of crude extract (based on filter mud)	4.19 ± 0.12 ^a^	4.75 ± 0.56 ^a^	2.25 ± 0.02 ^b^	1.55 ± 0.01 ^c^
Policosanol contents (mg/100 g)				
C22	16.87 ± 0.24 ^c^	64.92 ± 2.81 ^b^	22.17 ± 1.01 ^c^	164.66 ± 4.30 ^a^
C24	23.85 ± 5.24 ^c^	810.25 ± 49.74 ^a^	74.73 ± 0.24 ^c^	511.65 ± 12.61 ^b^
C26	8.47 ± 2.38 ^d^	1230.39 ± 35.64 ^b^	424.72 ± 7.31 ^c^	4103.30 ± 30.43 ^a^
C28	249.21 ± 1.40 ^d^	1430.11 ± 44.18 ^c^	2880.63 ± 19.29 ^b^	29,726.90 ± 62.85 ^a^
C30	102.72 ± 8.04 ^d^	1485.90 ± 24.44 ^c^	1897.28 ± 50.91 ^b^	17,233.82 ± 25.65 ^a^
C32	238.28 ± 4.85 ^c^	307.81 ± 0.69 ^c^	812.58 ± 20.74 ^b^	7167.79 ± 51.87 ^a^
C34	29.50 ± 2.22 ^d^	260.48 ± 5.39 ^b^	204.90 ± 5.21 ^c^	1148.10 ± 25.87 ^a^
Total	668.90 ± 10.92 ^d^	5589.87 ± 56.76 ^c^	6317.00 ± 77.16 ^b^	60,056.23 ± 73.50 ^a^
Phytosterol contents (mg/100 g)				
Campesterol	4482.19 ± 1.84 ^b^	4618.66 ± 36.82 ^a^	1600.01 ± 48.20 ^d^	3027.53 ± 7.87 ^c^
Stigmasterol	4423.49 ± 14.95 ^a^	3651.58 ± 13.06 ^b^	1201.00 ± 17.27 ^d^	2215.50 ± 2.60 ^c^
Beta-sitosterol	11,804.10 ± 7.09 ^a^	4478.10 ± 41.37 ^b^	1572.03 ± 11.44 ^d^	3020.79 ± 8.27 ^c^
Sitostanol	168.98 ± 6.47 ^d^	546.74 ± 16.15 ^b^	282.86 ± 20.82 ^c^	1734.35 ± 2.65 ^a^
Total	20,878.75 ± 17.41 ^a^	13,295.09 ± 24.67 ^b^	4655.90 ± 97.74 ^d^	9998.18 ± 16.09 ^c^

Each value represents the mean ± SD. Values with different superscript letters in the same row are significantly different (*p* < 0.05).

## Data Availability

The original data used to support the findings of this study are available from the first author and corresponding author upon request.

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
