# Peer review of "High Levels of Policosanols and Phytosterols from Sugar Mill Waste by Subcritical Liquefied Dimethyl Ether"

_foods, 2022, doi:10.3390/foods11192937_

Round 1

Reviewer 1 Report

This work describes the extraction of policosanols and phytosterols from sugarcane waste streams for use as nutraceuticals using sub-critical dimethyl ether. The work itself is very relevant in the context of increasing nutritional output from our crops – particularly from those parts we would normally consign to waste. The main work compares, and claims to build and improve on, previous work carried out by work using supercritical carbon dioxide for the same purpose. The work does indeed describe superior yields and provides, for the most part, a sound, referenced explanation for the increase in these yields.

Unfortunately, in terms of achieving the goals of what they have set out to do, the paper is seriously flawed beyond that. Most of this stems from the introduction which is riddled with mistakes, claims which are not supported by the references provided (some references are also missing) and data which is out of date. Specific notes are provided below, but most notably:

The authors have noted their superior yields with respect to scCO2 and have used this as their sole metric for their claims that this ‘study brought significant values for sugar mill waste using green extraction technology’. However, this claim is highly questionable given:

-        -The most promising yields were provided by carrying out a pre-treatment step using hexane, which one of their sources rightly states is ‘listed as No. 1 on the list of 189 hazardous air pollutants by the US Environmental Protection Agency’. It is also heavily restricted in the EU under REACh and listed as hazardous or banned on most Solvent Selection guides. E.g. https://pubs.rsc.org/en/content/articlehtml/2016/gc/c5gc01008j

This same guide also lists DME as hazardous due to the fact that, whilst it is indeed much more environmentally benign, it is far more hazardous to work with due to its extremely low boiling point and flash point.

-         -The authors correctly identify that supercritical fluid extraction is expensive due to the high pressures and temperatures involved and that subcritical fluid extraction allows for milder conditions. However, they neglect to account for the fact that DME is far, far more expensive than CO2. Linde’s suppliers, for instance, list a 34kg cylinder of CO2 at ~60 USD, whilst Sigma Aldrich lists an 8kg cylinder of DME as ~1200 USD! They also do not factor in other process metrics, such as PMI.

-         -The authors also incorrectly state that DME is listed as a safe extraction solvent for food in the EU – it is not, neither is hexane. Both have serious restrictions as to the specific areas that they can be used in. CO2 is, however. See Directive 2009/32/EC. This is perhaps the most significant one as it has serious implications about the potential to re-use these extracts in the food industry – especially with respect to scCO2 which is already approved of and used in the food industry.

As the actual scientific explanations for the data is sound, I will stop short of recommending this be rejected outright. However, this work cannot be considered for publication without serious revisions. Most notably:

-       -  If the authors wish to state that they are using green extraction technology, they must be able to clearly explain and justify why they are using 2 (hexane, DME) notoriously hazardous and restricted solvents given the availability of other, far more benign and sustainable solvents out there and the range of literature that has been available on solvent selection for nearly 10 years to help make these substitutions.

-       -  In a similar vein, the authors must be able to demonstrate, beyond just yields, that their work has successfully built upon previous work in the context they are discussing.

-       -  The introduction must make sure that the references being cited actually support the claims being made.

Other general notes:

Reference 2, the most heavily cited paper in this study: Asstard et al. The author’s name is Attard, not Asstard – a very insulting and derogatory term in English. https://en.wiktionary.org/wiki/asstard

Soxhlet is named after Franz von Soxhlet, so Soxhlet should always be spelled with capital S

Specific notes:

Lines 37&38 ‘In 2017, the FAO listed 102 countries that cultivated sugarcane (Saccharum officinarum 37 L.), with land area of 26 million hectares (Mha), producing fresh sugarcane of 1,842 million 38 tons (Mt) annually [1].’

This reference does not appear to support this claim. If this is from the FAO database, it would be best to get the figures direct from the database and cite it directly.

Lines 39-42 ‘Harvesting of sugarcane crops and processing at sugar mills generate significant amounts of by-products such as sugarcane leaves, bagasse and filter mud (filter cake) or pressed mud. Sugarcane leaves are agricultural waste from the harvesting step, whereas bagasse and filter mud are discarded during the milling process [2]’

This reference does not directly support the claim – it makes no assessment of how waste is generated, only on how to utilise it. It does cite other work where this has been looked at (Eggleston 2014, Panday 2000, Zuurbier 2008), so if these papers do show this assessment it would be better to cite these directly.

Lines 52-53 – “Policosanols comprise a mixture of long-chain aliphatic alcohols with chain lengths ranging from 22 to 34 carbons”.

Only one policosanol – octocosanol – is discussed in the literature cited. Do the same benefits apply to the other 6 in this bracket?

Line 58 – “with more than 100 types of phytosterols reported in plant species [10,11].”

Reference 10 talks about phytosterols generally, but does not give a figure for how many were looked at or have been identified. Reference 11 counts just 44.

Lines 62-63 “Advantages of solvent extraction include convenience, ease of use and hassle-free, while disadvantages are the remaining chemical residues in the final product that are toxic to the environment [14-16]”

References 14 and 15 do not discuss solvent extraction at all. Reference 16 does – however since it, rightly, also points at that hexane is “listed as No. 1 on the list of 189 hazardous air pollutants by the US Environmental Protection Agency” – it raises the very serious question as to why it is being used so extensively in this study where the aim is to develop a green processing method. It also does not discuss, or quantify solvent residues in the final product.

Lines 66-67 “However, the use of high-pressure supercritical fluid technology is limited due to the cost [17, 14].”

Both of these references discuss ultrasound treatment, not supercritical fluid technology.

Lines 78-79 “Several solvents can be used as subcritical liquids such as water, propane, butane, dimethyl ether (DME), and 1,1,1,2‐tetrafluoroethane [19, 20]”

Reference 19 only discusses water and reference 20 only discusses butane. In addition, to discuss the previous lines about this being a new technique – subcritical water is hardly new. It is heating the water above 100 oC whilst maintaining it in liquid form – in other words pressure cooking! In scientific fields, the first paper discussing it as an extractant dates back to 1962 and by 2000 had over 100 papers discussing it!

Lines 80-81 “DME has been authorized by the European Food Safety Authority as a safe extraction solvent to produce foodstuffs and food ingredients [21]”

This reference is an opinion piece, not formal authorisation.  The current, formal European legislation on food grade solvents lists only 7 for general use: Propane, Butane, Ethyl acetate, Ethanol, Carbon dioxide, Acetone and Nitrous oxide. DME is authorised for only one usage: Preparation of defatted animal protein products – see Directive 2009/32/EC

Lines 81-83 – “It is not a greenhouse gas and does not cause ozone depletion. Moreover, the global warming potential (GWP) index of DME is one order of magnitude lower than that of CO2 [22,23]

The first sentence is directly copied from reference 22 – which provides no reference itself to back up that claim. Reference 23 does not include DME as one of the solvents it is assessing the GWP of. It is also over 30 years old and our knowledge of the atmospheric impact of VOCs has advanced considerably in this time.

Lines 156-157 – “The example of GC-MS chromatograms of sample and standard policosanols were shown in Figure.1-2.S”

This is being discussed directly, therefore should be in the main manuscript, not the supplementary information.

Line 229 – I have no idea what reference 35 is meant to be adding here. Cannot comment on reference 36 as it is not in English, but as the text is discussing basic principles anyway, I don’t see why it is needed.

Lines 232-237 - Previous studies reported similar trends regarding the effect of particle size on extraction yields and antioxidant activity of peanut skin. Particle size of 425 μm yielded the highest oil yield compared to higher particle size of 500 μm and smaller particle size of 355 μm [37]. Decrease in particle size resulted in greater surface area of particles in contact with the solvent that enhanced the leaching of active compounds due to increased mass transfer. Moreover, decrease in particle size and gradual increase in temperature also positively affected the extraction rate [38, 39].

This is standard undergraduate solvation chemistry. It is questionable whether something so trivial merits discussion in an IF> 5 journal.

The DOI for references 4 and 26 do not match the references provided.

The DOI for reference 10 links to reference 8

The DOI for reference 55 links to reference 3

The DOI for references 34 and 41 are broken

Author Response

high for application in industries.

-DME is a colorless gas with a slight ether-like fragrance at room temperature and pressure. Due to some special properties, such as the strong ability for extracting organic compounds and water, high extraction rate, cheap price, low extraction temperature, and energy consumption, environmental friendliness, safety, and good compressibility, the application of liquefied DME to the extraction process shows many advantages and has strong potential market competitiveness (Zhenga & Watanab ,2022).

Zheng, Q.; Watanabe, M. Advances in Low-temperature Extraction of Natural Resources Using  Liquefied Dimethyl Ether. Resources Chemicals and Materials. (2022), 1(1), 16-26.

- Dimethyl ether (DME) is a versatile raw material and an interesting alternative fuel that can be produced by the catalytic direct hydrogenation of CO2 which this process has attracted the attention of the industry due to the environmental benefits of CO2 elimination from the atmosphere and its lower operating costs. (Mota et al., 2021; Olah et al., 2009; Catizzone et al., 2018).

-Mota also reported DME can be synthesized by CO2 and is one strategy to add the value to CO2 as green house gas.

Mota, N., Ordoñez, E.M., Pawelec, B., Fierro, J.L.G., Navarro, R.M. Direct Synthesis of Dimethyl Ether from CO2: Recent Advances in Bifunctional/Hybrid Catalytic Systems. 2021 Catalysts, 11, 411. https://doi.org/10.3390/catal11040411

-The authors also incorrectly state that DME is listed as a safe extraction solvent for food in the EU – it is not, neither is hexane. Both have serious restrictions as to the specific areas that they can be used in. CO2 is, however. See Directive 2009/32/EC. This is perhaps the most significant one as it has serious implications about the potential to re-use these extracts in the food industry – especially with respect to scCO2 which is already approved of and used in the food industry.

Response: Thanks for the comment.

The Generally Recognized as Safe (GRAS) status of dimethyl ether is corroborated through the independent safety evaluations conducted by European Food Safety Authority (EFSA) and Food Standards Australia New Zealand (FSANZ) who approved the use of dimethyl ether as an extraction solvent.

In EFSA, dimethyl ether is authorized for the food preparation of defatted animal protein products (EU Directive 2009/32/EC) and FSANZ was submitted as part of the applications to processing aid in the production of dairy and non-dairy food ingredients and products (FSANZ, 2011).

Dimethyl ether, therefore, may be marketed and sold for its intended purpose in the U.S. without the promulgation of a food additive regulation under Title 21, Section 170.3 of the Code of Federal Regulations.

https://efsa.onlinelibrary.wiley.com/doi/pdf/10.2903/j.efsa.2009.984.

https://efsa.onlinelibrary.wiley.com/doi/pdf/10.2903/j.efsa.2015.4174

https://www.fda.gov/media/113335/download

As the actual scientific explanations for the data is sound, I will stop short of recommending this be rejected outright. However, this work cannot be considered for publication without serious revisions. Most notably:

 - If the authors wish to state that they are using green extraction technology, they must be able to clearly explain and justify why they are using 2 (hexane, DME) notoriously hazardous and restricted solvents given the availability of other, far more benign and sustainable solvents out there and the range of literature that has been available on solvent selection for nearly 10 years to help make these substitutions.

Response: Thanks. To make it clear the authors revised as in line 84-89

- As mentioned previously, the authors further used hexane as sample pretreatment just to increase the policosanol and phytosterol contents and the authors do not intend to claim that the technique application of hexane as the pretreatment step before using SUBLDME technology is the green technology. Since the authors directly used the subcritical dimethyl ether technique without sample pretreatment by hexane to extract the crude extract from the by-product samples and found that it gave high yield policosanol and phytosterol content (as in Table 3). The authors intend to use “green technique” only for SUBLDME

- DME is regarded to be safe for the human body and can be allowed to be applied in the food industry in some countries. It has been approved as a safe extraction solvent for the production of foodstuffs and food ingredients by the European Food Safety Authority (EFSA), by the Food Standards Australia New Zealand (FSANZ) and by the U. S. Food and Drug Administration

- In a similar vein, the authors must be able to demonstrate, beyond just yields, that their work has successfully built upon previous work in the context they are discussing.

Response: Thanks. The authors did discuss the data of yield of crude extract by the direct SUBLDME method and quantities of policosanols and phytosterol with the supercritical method by Attard et al (2015) since the their study used the same material as this study. As in line 422-584

The introduction must make sure that the references being cited actually support the claims being made.

Other general notes:

Reference 2, the most heavily cited paper in this study: Asstard et al. The author’s name is Attard, not Asstard – a very insulting and derogatory term in English. https://en.wiktionary.org/wiki/asstard

Response: Thank you for your suggestion. It is very big mistake. The authors revised the name as your suggestion.

 In line#71, 416, 461, 483, 488, 505, 560

Soxhlet is named after Franz von Soxhlet, so Soxhlet should always be spelled with capital S

Response: Thanks. The authors revised as your suggestion.

Line#260, 406,412,  414, 424,647,

Specific notes:

Lines 37&38 ‘In 2017, the FAO listed 102 countries that cultivated sugarcane (Saccharum officinarum 37 L.), with land area of 26 million hectares (Mha), producing fresh sugarcane of 1,842 million 38 tons (Mt) annually [1].’

This reference does not appear to support this claim. If this is from the FAO database, it would be best to get the figures direct from the database and cite it directly.

Response: Thanks. The authors revised Ref 1 to FAO as the suggestion in Line 42-43

FAOSTAT, 2001. Food and Agricultural Organization of the United Nations. Crop Statistics        Data Base.

Lines 39-42 ‘Harvesting of sugarcane crops and processing at sugar mills generate significant amounts of by-products such as sugarcane leaves, bagasse and filter mud (filter cake) or pressed mud. Sugarcane leaves are agricultural waste from the harvesting step, whereas bagasse and filter mud are discarded during the milling process [2]’

This reference does not directly support the claim – it makes no assessment of how waste is generated, only on how to utilise it. It does cite other work where this has been looked at (Eggleston 2014, Panday 2000, Zuurbier 2008), so if these papers do show this assessment it would be better to cite these directly.

Response: Thanks.

The author revised the reference citation from Attard et al., 2015 to Eggleston et al., 2014 as in Line 47

Lines 52-53 – “Policosanols comprise a mixture of long-chain aliphatic alcohols with chain lengths ranging from 22 to 34 carbons”.

Only one policosanol – octocosanol – is discussed in the literature cited. Do the same benefits apply to the other 6 in this bracket?

Response: Thanks. The reference was added as in line 58 and in reference list (Lukashevich, Davidson, Moreines, & Berlin, 2006)

[7]Lukashevich, V., Davidson, M. H., Moreines, J., & Berlin, R. G. (2006). Beeswax policosanol     failed to demonstrate lipid-altering effects in well-controlled clinical trials.

There are many review articles on policosanol benefits. To see the health benefit, some study use octacosanol as sample while other used C30 (Weerawatanakorn, et al., 2019) or as mixture extract of policosanol.

Weerawatanakorn, M.; Meerod, K.; Wongwaiwech, D.; Ho, C. T. Policosanols: chemistry, occurrence, and health effects. Current Pharmacology Reports. 2019, 5(3), 131-149.

- Policosanol has been used as in functional foods because it has multiple health improving properties, including the lowering of lipids, anti-aging, cardiovascular protection against diabetes, hypercholesterolemia, Parkinson’s disease, inflammation, ulcers and cancer. (Shen et al., 2019)

Shen, J., Luo, F., & Lin, Q. (2019). Policosanol: Extraction and biological functions. Journal of       Functional Foods, 57, 351-360. https://doi.org/10.1016/j.jff.2019.04.024

Line 58 – “with more than 100 types of phytosterols reported in plant species [10,11].”

Reference 10 talks about phytosterols generally, but does not give a figure for how many were looked at or have been identified. Reference 11 counts just 44.

Response: Thanks. The author revised the reference citation from [10,11] to Moreau et al., 2002 and the claim was changed to ‘’ more than 200 different types of phytosterols have been reported in plant species’’as in line 62

Reference:

[10] Moreau, R. A., Whitaker, B. D., & Hicks, K. B. (2002). Phytosterols, phytostanols, and their conjugates in foods: structural diversity, quantitative analysis, and health-promoting uses. Progress in lipid research, 41(6), 457-500. https://doi.org/10.1016/s0163-  7827(02)00006-1

Lines 62-63 “Advantages of solvent extraction include convenience, ease of use and hassle-free, while disadvantages are the remaining chemical residues in the final product that are toxic to the environment [14-16]”

References 14 and 15 do not discuss solvent extraction at all. Reference 16 does – however since it, rightly, also points at that hexane is “listed as No. 1 on the list of 189 hazardous air pollutants by the US Environmental Protection Agency” – it raises the very serious question as to why it is being used so extensively in this study where the aim is to develop a green processing method. It also does not discuss, or quantify solvent residues in the final product.

Response: Thanks. The authors removed the reference citation of 14,15.  As mentioned previously, Since the authors directly used the subcritical dimethyl ether technique without sample pretreatment by hexane to extract the crude extract from the by-product samples and found that it gave high yield policosanol and phytosterol content (as in Table 3). The authors intend to use “green technology” only for SUBLDME, but not the extraction processing method of sample pretreatment.

Lines 66-67 “However, the use of high-pressure supercritical fluid technology is limited due to the cost [17, 14].”

Both of these references discuss ultrasound treatment, not supercritical fluid technology.

Response: Thanks. The authors removed the reference citation at [17, 14] to Sahena et al., 2009 as in line 71. Since this reference does support the claim that the pressure of supercritical fluid technology is high and the equipment is expensive.

Reference:  Sahena, F., Zaidul, I., Jinap, S., Karim, A. A., Abbas, K. A., Norulaini, N., & Omar, A. (2009).             Application of supercritical CO2 in lipid extraction - A review. Journal of Food       Engineering, 95, 240– 253. https://doi.org/10.1016/j.jfoodeng.2009.06.026

Lines 78-79 “Several solvents can be used as subcritical liquids such as water, propane, butane, dimethyl ether (DME), and 1,1,1,2‐tetrafluoroethane [19, 20]”

Reference 19 only discusses water and reference 20 only discusses butane. In addition, to discuss the previous lines about this being a new technique – subcritical water is hardly new. It is heating the water above 100 oC whilst maintaining it in liquid form – in other words pressure cooking! In scientific fields, the first paper discussing it as an extractant dates back to 1962 and by 2000 had over 100 papers discussing it!

Response: Thanks. The reference on propane was added as [19]

Reference: [19] Teixeira, G. L., Ghazani, S. M., Corazza, M. L., Marangoni, A. G., & Ribani, R. H. (2018).             Assessment of subcritical propane, supercritical CO2 and Soxhlet extraction of oil from   sapucaia (Lecythis pisonis) nuts. The Journal of Supercritical Fluids, 133, 122-132.             https://doi.org/10.1016/j.supflu.2017.10.003

Lines 80-81 “DME has been authorized by the European Food Safety Authority as a safe extraction solvent to produce foodstuffs and food ingredients [21]”

This reference is an opinion piece, not formal authorisation.  The current, formal European legislation on food grade solvents lists only 7 for general use: Propane, Butane, Ethyl acetate, Ethanol, Carbon dioxide, Acetone and Nitrous oxide. DME is authorised for only one usage: Preparation of defatted animal protein products – see Directive 2009/32/EC

Response: Thanks. The authors revised the sentence following Directive 2009/32/EC As suggestion as in line 84-89.

Lines 81-83 – “It is not a greenhouse gas and does not cause ozone depletion. Moreover, the global warming potential (GWP) index of DME is one order of magnitude lower than that of CO2 [22,23]

The first sentence is directly copied from reference 22 – which provides no reference itself to back up that claim. Reference 23 does not include DME as one of the solvents it is assessing the GWP of. It is also over 30 years old and our knowledge of the atmospheric impact of VOCs has advanced considerably in this time.

Response: Thanks. The authors revised the sentence to “It is not a greenhouse gas and does not deteriorate ozone layer”as in line 89. The authors also removed “Moreover, the global warming potential (GWP) index of DME is one order of magnitude lower than that of CO2 [22,23]” and added the 2 references below

[22]Subratti A, Lalgee LJ, Jalsa NK. Liquifed dimethyl ether (DME): a green solvent for the extraction of hemp (Cannabis sativa L.) seed oil. Sustain Chem Pharm. 2019;12:100144

[23]Subratti A, Lalgee LJ, Jalsa NK. Effecient extraction of black cumin (Nigella sativa L.) seed oil containing thymol, using liquefed dimethyl ether (DME). J Food Process Preserv. 2019;43(4):e13913

Lines 156-157 – “The example of GC-MS chromatograms of sample and standard policosanols were shown in Figure.1-2.S”

This is being discussed directly, therefore should be in the main manuscript, not the supplementary information.

Response: Thanks. The authors put the chromatograms in the manuscript as the suggestion as in line 409 and 480 as Figure 2-5

Line 229 – I have no idea what reference 35 is meant to be adding here. Cannot comment on reference 36 as it is not in English, but as the text is discussing basic principles anyway, I don’t see why it is needed.

Response: Thanks. The authors removed the reference citation [35] and added the reference citation as Prasedya et al., 2021; Yatma et al., 2022 and Belwal et al., 2016 as in line 391-392

Reference: [35-38]

[35]Prasedya, E. S., Frediansyah, A., Martyasari, N. W. R., Ilhami, B. K., Abidin, A. S., Padmi, H.,& Sunarwidhi, A. L. (2021). Effect of particle size on phytochemical composition and antioxidant properties of Sargassum cristaefolium ethanol extract. Scientific       Reports11(1), 1-9. https://doi.org/10.1038%2Fs41598-021-95769-y

[36] Yatma, F., Falah, S., Ambarsari, L., Aisyah, S. I., & Nurcholis, W. (2022). Optimization of Extraction of Phenolic and Antioxidant Activities from Celosia cristata Seeds Using Response Surface Methodology. Biointerface Research in Applied Chemistry, 13(2), 2069-5837. https://doi.org/10.33263/BRIAC132.148

[37]Belwal, T., Dhyani, P., Bhatt, I. D., Rawal, R. S., & Pande, V. (2016). Optimization extraction conditions for improving phenolic content and antioxidant activity in Berberis asiatica            fruits using response surface methodology (RSM). Food chemistry, 207, 115-124. http://dx.doi.org/10.1016/j.foodchem.2016.03.081

Lines 232-237 - Previous studies reported similar trends regarding the effect of particle size on extraction yields and antioxidant activity of peanut skin. Particle size of 425 μm yielded the highest oil yield compared to higher particle size of 500 μm and smaller particle size of 355 μm [37]. Decrease in particle size resulted in greater surface area of particles in contact with the solvent that enhanced the leaching of active compounds due to increased mass transfer. Moreover, decrease in particle size and gradual increase in temperature also positively affected the extraction rate [38, 39].

This is standard undergraduate solvation chemistry. It is questionable whether something so trivial merits discussion in an IF> 5 journal.

Thanks.

The DOI for references 4 and 26 do not match the references provided.

Response: Thanks. The authors revised as your suggestion.

The DOI for reference 10 links to reference 8

Response: Thanks. It was revised.

The DOI for reference 55 links to reference 3

Response: Thanks. It was revised.

The DOI for references 34 and 41 are broken

Response: Thanks. It was revised

Reviewer 2 Report

Why have the authors not included the discussion on morphological changes of the samples before and after treatment? 

In general, the manuscript was prepared well and this study shows good scientific quality. However, it is more appropriate if the authors could include the morphological changes of the material/sample before and after extraction treatment. The results can help confirm certain conditions, such as the breakage of the cell wall and correlate the yield findings by observing the cell wall disruption due to the treatment.

Author Response

Response: Thanks. The authors added Figure as the suggestion in supplementary data as Figure 1 S and we added this result in line 292-298.

Reviewer 3 Report

The authors previously demonstrated that the SUBLDME extraction can be applied for the recovery of policosanols. Therefore, in this study, they investigated a different material.

The entire manuscript should be very carefully checked, since it contains numerous mistakes: technical and grammar, even some terms, such as " defatted rice bran", most probably from the previous manuscript. Therefore, careful revision is necessary. Furthermore, the abstract and introduction are not written in a clear way, making it unclear what exactly was investigated, it is very general and lacks specific details.

My recommendation is to include an explanation into the Introduction section with more detail on what was explored; for example, the impact of the sieving mesh size and extraction temperature on extraction yield and content of bioactive compounds, as well as pretreatment, etc. Moreover, the Introduction section should highlight the innovative aspect of this manuscript and what has been improved, and the difference from previous the study since it cannot be just a new source material. 

Also, the abstract is not clearly written as well. Please rephrase it with including more details on what was the goal of this study and what was investigated. 

Please rephrase:" Further process of bagasse, filter mud, and sugarcane leaves discarded as sugar mill by-products by extracting nutraceuticals of high value is crucial for the development of a sustainable bio-economy." Additionally, the entire manuscript contains sentences that are unclear due to their structure.

"Subcritical liquefied extraction (SUBLE) is a new separation technology developed after supercritical fluid extraction (SFE) and both are the pressurized low temperature technology with similar advantages ". In the subcritical processes, if water is used as a solvent, extraction temperature is not low, so please rephrase it. 

"Compared to SFE, the SUBLE technique requires lower pressure and costs less energy and investment, which makes it less expensive and more practical for large-scale applications."  Supercritical fluid extraction is more implemented in industries, therefore, please rephrase it. Also, include a reference for this statement, that subcritical is less expensive than supercritical fluid extraction. 

A preliminary study also indicated that there are high contents of phytosterols extracted from sugarcane leaves by SUBLDME extraction.” Please include a reference for this statement.

Please describe in detail the specifications (min/max pressure, capacity, producer…) of the lab-scale equipment used for extraction. Including a figure only is not enough. 

Title of section 2.6. Effect of sample pretreatment on policosanol and phytosterol contents sounds as title of section from Results and discussion. Please adjust it, maybe to just “Sample pretreatmentwithout “effect”.

How were the process conditions chosen? Based on preliminary experiments or literature? If it is literature, please include a reference. Why were only two temperatures applied? 

Higher applied temperature was more effective, however, an even higher temperature can be even more effective. 

Section Effect of particle size and extraction temperature on yield and phytosterol contents by SUBLDME extraction does not possess any significant explanations regarding the influence of temperature. 

Author Response

esponse to Reviewer 3 Comments

Comments and Suggestions for Authors Reviewer 3

The authors previously demonstrated that the SUBLDME extraction can be applied for the recovery of policosanols. Therefore, in this study, they investigated a different material.

The entire manuscript should be very carefully checked, since it contains numerous mistakes: technical and grammar, even some terms, such as " defatted rice bran", most probably from the previous manuscript. Therefore, careful revision is necessary. Furthermore, the abstract and introduction are not written in a clear way, making it unclear what exactly was investigated, it is very general and lacks specific details.

My recommendation is to include an explanation into the Introduction section with more detail on what was explored; for example, the impact of the sieving mesh size and extraction temperature on extraction yield and content of bioactive compounds, as well as pretreatment, etc. Moreover, the Introduction section should highlight the innovative aspect of this manuscript and what has been improved, and the difference from previous the study since it cannot be just a new source material.

Response: Thanks for comment. The authors revised the introduction as the suggestion in Line 227-239

Also, the abstract is not clearly written as well. Please rephrase it with including more details on what was the goal of this study and what was investigated.

Response: Thanks for comment. The authors added information on goal as suggestion in Line 24-28

Please rephrase:" Further process of bagasse, filter mud, and sugarcane leaves discarded as sugar mill by-products by extracting nutraceuticals of high value is crucial for the development of a sustainable bio-economy." Additionally, the entire manuscript contains sentences that are unclear due to their structure.

Response: Thanks for comment. The authors rephrase as suggestion in Line 21-23

“By extracting nutraceuticals with high value from bagasse, filter mud, and sugarcane leaves discarded as sugar mill by-products is crucial for the development of a sustainable bio-economy”

"Subcritical liquefied extraction (SUBLE) is a new separation technology developed after supercritical fluid extraction (SFE) and both are the pressurized low temperature technology with similar advantages ". In the subcritical processes, if water is used as a solvent, extraction temperature is not low, so please rephrase it.

Response: Thanks for comment. The authors rephrase as suggestion in Line 77-79

"Compared to SFE, the SUBLE technique requires lower pressure and costs less energy and investment, which makes it less expensive and more practical for large-scale applications."  Supercritical fluid extraction is more implemented in industries, therefore, please rephrase it. Also, include a reference for this statement, that subcritical is less expensive than supercritical fluid extraction.

Response: Thanks for comment. The authors rephrase as suggestion in Line 81-83

“A preliminary study also indicated that there 

Round 2

Reviewer 1 Report

The revised manuscript is much improved. However some of the edits have led to some repetition of details or a couple of confusing statements. The document should undergo acceptance of the changes so it can be properly proof-read prior to publication.

Author Response

Reviewer 1

Comments and Suggestions for Authors

The revised manuscript is much improved. However some of the edits have led to some repetition of details or a couple of confusing statements. The document should undergo acceptance of the changes so it can be properly proof-read prior to publication.

Response: Thanks. The authors did as the reviewer suggestion by acceptance of the changes and made some changes as in line 40,72,84,86,108,254,268,293,306,307,378, 379,392,394,444,

Some reference list also were revised.

Reviewer 3 Report

The authors revised and corrected the Manuscript according to most of the suggestions. However, the manuscript still contains numerous technical and grammar mistakes.

Author Response

Reviewer 2

Comments and Suggestions for Authors

The authors revised and corrected the Manuscript according to most of the suggestions. However, the manuscript still contains numerous technical and grammar mistakes.

Response: Thanks. The authors check the gramma and make some change at line 23-24, 39,53, 86, 89, 101, 107,132,239,241,243,244, 246, 252, 257,

Some reference list also were revised.
